# Biodiversity, Biochemical Profiling, and Pharmaco-Commercial Applications of *Withania somnifera*: A Review

**DOI:** 10.3390/molecules28031208

**Published:** 2023-01-26

**Authors:** Harshita Gaurav, Divyanshu Yadav, Ankita Maurya, Himanshu Yadav, Ramanand Yadav, Amritesh Chandra Shukla, Minaxi Sharma, Vijai Kumar Gupta, Javier Palazon

**Affiliations:** 1Department of Botany, University of Lucknow, Lucknow 226007, India; 2Haute Ecole Provinciale de Hainaut-Condorcet, 11 Rue de la Sucrerie, 7800 Ath, Belgium; 3Biorefining and Advanced Materials Research Center, SRUC, Kings Buildings, West Mains Road, Edinburgh EH9 3JG, UK; 4Department of Biotechnology, Graphic Era Deemed to Be University, Dehradun 248002, India; 5Department of Biology, Healthcare and the Environment, University of Barcelona, 08028 Barcelona, Spain

**Keywords:** *Withania somnifera*, ashwagandha, traditional use, botany, phytochemistry, pharmacological and biological activity

## Abstract

*Withania somnifera* L. Dunal (Ashwagandha), a key medicinal plant native to India, is used globally to manage various ailments. This review focuses on the traditional uses, botany, phytochemistry, and pharmacological advances of its plant-derived constituents. It has been reported that at least 62 crucial and 48 inferior primary and secondary metabolites are present in the *W. somnifera* leaves, and 29 among these found in its roots and leaves are chiefly steroidal compounds, steroidal lactones, alkaloids, amino acids, etc. In addition, the whole shrub parts possess various medicinal activities such as anti-leukotriene, antineoplastic, analgesic, anti-oxidant, immunostimulatory, and rejuvenating properties, mainly observed by in vitro demonstration. However, the course of its medical use remains unknown. This review provides a comprehensive understanding of *W. somnifera*, which will be useful for mechanism studies and potential medical applications of *W. somnifera*, as well as for the development of a rational quality control system for *W. somnifera* as a therapeutic material in the future.

## 1. Introduction

The World Health Organization (WHO) stated that approximately 30% of drugs contain plant-derived compounds, and approximately more than half of the global population depends upon plant-derived medicines, including those of Ayurveda, the traditional plant-based medicine of India [1,2]. The Botanical Survey of India (BSI) reported that approximately 7500 of the 15,000 plant species in India are used for medicinal purposes [3]. A list of 32 therapeutically important plants of India prioritized by the National Bank for Agriculture and Rural Development (NABARD) includes *Withania somnifera*, regularly called as ashwagandha. The National Medicinal Plant Board (NMPB), India, has also described *Withania* species as a valuable therapeutic plant in great demand in domiciliary and global markets [4,5].

*W. somnifera*, indigenous to the Indian sub-continent, is used as indigenous medicine throughout Southeast Asia. Its roots, seeds, and leaves have been used for over 3000 years for multiple health purposes, and are included in Ayurvedic, Allopathic, Unani, homeopathic, and other medical systems [6]. Due to its superlative therapeutic properties, it is also known as the Queen of Ayurveda or a Rasayana herb. *Withania* spp. are members of the Solanaceae family (Bentham and Hooker, 1862–1883), also known as the deadly nightshade family. The Solanaceae family embraces 84 genera, comprising 3000 species, distributed all over the globe [6]. The genus *Withania* contains 26 species, only 2 of which, W. *somnifera* and *W. coagulans* (known as Paneer booti, Ashutosh booti or rishyagandha), are economically and medicinally important in many regions [7]. Other potentially important species are *W. simonii*, *W*. *adunensis*, and *W. riebeckii.*

Two subspecies have also been reported—*W. somnifera Dunal* and *W. somnifera Kaul* [8]. Both species of *Withania*, i.e., *W. somnifera* and *W. coagulans*, are cultivated on a large scale in India [9]. In Sri Lanka, two cultivars of *W. somnifera* are used, but the Indian cultivar is more suitable for drug development due to its starchy nature, whereas the local cultivar has fibrous roots that are difficult to transform into powder [10]. Its common name, ashwagandha, is a combination of two Sanskrit words, ashwa and gandha, and means “horse’s smell”, referring to the sweaty horse-like smell of the roots [9,11]. Other names include winter cherry, poison gooseberry [12,13], or “Indian ginseng”, due to its restorative properties.

However, the mechanism of drug use by *W. somnifera* has not been elucidated. Accordingly, there are unreasonable points in the current *W. somnifera* quality management system. In fact, its medicinal value has received less attention, despite its long history of clinical application. In this review, information regarding various aspects of *W. somnifera* was collected from peer-reviewed journals for the period 2000–2022. PubMed, Shod Ganga, Web of Science, Google Scholar, etc., was used to retrieve electronic information. Based on this information, we performed a full review of *W. somnifera* aiming to provide information to better understand the pharmacological mechanism and potential medicinal applications of this species, and to develop a reasonable quality control system for *W. somnifera* as medicinal material in the future.

## 2. Biodiversity of *W. somnifera*

*W. somnifera* is a drought-resistant, small, perennial shrub 30–150 cm in height (Figure 1). It is cultivated throughout the dry tropical parts of Afghanistan, Baluchistan, Canary Islands, China, Congo, Egypt, India, Iran, Israel, Jordan, Madagascar, Morocco, Nepal, Pakistan, South and East Africa, Spain, Sri Lanka, Sudan, and Yemen [14].

According to the biological classification, Withania belongs to the kingdom Plantae (plants), the sub-kingdom Tracheophytes (vascular plants), division Angiospermae, class Eudicots, clade Asterids, order Solanales, family Solanaceae, sub-family Solanoideae, tribe Physaleae, genus *Withania*, and species *somnifera* (2n = 48), *adpressa*, *coagulans*, *frutescens* [5,7,15,16,17] (Table 1).

According to Charaka, one of the main donors of Ayurveda, *W. somnifera* has miraculous potential. A person treated with this plant “can live for longer, regain youth, have a sharper memory and intellect and freedom from diseases, and acquire a lustrous complexion and horse-like vigor’.

### 2.1. Leaves

The aerial part of the plant is covered with minute trichomes. The branches are round and leaves are up to 30–80 × 20–50 mm, narrowed into 5–20 mm long petioles. Morphologically, leaves are simple, exstipulate, oblique, petiolate, broadly ovate to oblong, entire, shiny smooth, dull green in color, globose, and elliptical. Leaves are alternate on vegetative shoots and opposite on floral shoots, having margins entirely to slightly wavy. The leaves are almost hairless above, densely hairy below, and bitter in taste. Its leaves have many advantages as shown in Table 2 [5,27].

### 2.2. Roots

Roots are long tap roots, 15–25 cm in length, stout and fleshy, tapering, and brownish-white or light-yellow-colored root with a horse-like smell and bitter taste. Its roots possess many valuable biological activities as shown in Table 3 [5,27].

### 2.3. Flowers and Fruits

Flowers are inconspicuous, solitary, green or dull yellow in color, 4–6 mm in diameter, bisexual, pedicellate, highly protogynous, small, and bell-shaped. Cross-pollination is carried out by ants and bees [32]. Inflorescences are axillary umbellate cymes. The calyx is gamosepalous with valvate aestivation. The corolla is greenish-yellow, valvate, campanulate, and has 5–8 mm lobes. Stamens (5) are polyandrous and epipetalous. Anthers are dithecous, basifixed, and introse. The ovary is superior, bilocular, and has axile placentation. Fruit are orange-red berries containing very light seeds, which are small, smooth, lens-shaped, or kidney-shaped, 2–2.5 mm length × 1.5–2 mm width × 0.5 mm height. Its flowers and fruits are having many phytoconstituents which helps in anti-microbial activities as well as works as a milk thickening agents as shown in Table 4 [33].

### 2.4. Ayurvedic Properties

*W. somnifera* has played a role in human society since prehistoric times. Its medicinal nature is specified in Ayurvedic books, such as *Astanga Hridaya*, *Bhava Prakasha Nighantu*, *Charaka Samhita*, and *Sushruta Samhita* [12,28,37]. Its Ayurvedic properties are Atishukrala, boosts semen quality and quantity; Balya, which provides power; Vajikara (andrology), which enhances sensual problems; Kshayapaha, which helps to treat weakness and provides nutritive conditions; Rasayani (chemotaxis), which rejuvenates the body and removes toxins; Shwitrapaha, which heals white discoloration of the skin; and Shothahara, which treats swelling in tissue and clears impurities (Ama). Some Ayurvedic qualities includes Rasa (taste) Tikta (bitter), Kasaya (astringent), Madhura (sweet), katu (pungent), Guna (characteristics), Snigdha (Softness, unctuous), Laghu (light), Virya (potency), Usna (hot potency), Vipaka-Madhura and Doshakarma-Kapha (natural coolants), Vata, and Samaka (pain management) [38].

## 3. Biochemical Profiling of *W. somnifera*

*Withania somnifera* contain many important phyto-chemicals that perform various antimicrobial actions against pathogens. Different metabolic constituents such as secondary metabolites may differ with the tissue type, growth conditions, and variety of *W. somnifera*. These variations leading to unpredictable medication and health-promoting properties of various commercially available preparations developed with the help of *Withania* spp. [39]. In a multi-component therapeutic system, this makes standardization of the composition of herbal preparations and commercial exploitation of this plant difficult, because different components can influence health effects through complex multi-target interactions.

It has been reported that 62 major minor primary and secondary metabolites are present in the leaves, and 48 are present in roots, 29 of which are found in both the leaves and roots [40]. Among the bio-active constituents of *Withania* somnifera reported in the literature are steroidal flavanol glycosides, glycowithanolides, steroidal lactones, and phenolics as shown in Table 5 and Figure 2 [40,41].

As per the literature, 12 alkaloids, 40 withanoloids, and several sitoindosides have been isolated from the *Withania* species [48]. Several amino acids, alanine, aspartic acid, glycine, glutamic acid, cysteine, tyrosine, chlorogenic acid, glycosides, glucose, condensed tannins, iron, and flavonoids have also been reported [49].

Therefore, these phytochemicals have been reported in the well-explored species *W. coagulans* and *W*. *somnifera*. In contrast, few studies have explored *W. aristate*, *W. obtusifolia*, *W. adpressa*, and *W. frutescens*, reporting antimicrobial [32], diuretic [50], anti-proliferative cytotoxic [22], and antifungal [51] properties. Phytochemicals in leaves such as withanolides, 14α, 15α, 17β, 22R withanolide F, and withanolide J have been evaluated, but other plant parts have not yet been studied.

### 3.1. Withanolides

Withanolides are unique classes of steroidal lactones. Over 130 withanolides are present in 15 genera of the Solanaceae family. As per some studies, withanolides are isolated from some other Leguminosae and Labiatae families, as well as from some marine organisms [5].

*W*. *somnifera* plants fabricate the largest number of withanolides comprising highly diversified functional group. Withanolides are key biochemicals, ranging from 0.001% to 0.5% dry weight concentration in the leaves and roots of *Withania somnifera.* Withanolides are a 28-carbon-containing natural steroidal lactone built on an ergostane skeleton, in which C-22 and C-26 are oxidized to form six- or five-membered lactone rings. Withanolides are also known as 22-hydroxy ergostan-26-oic acid-26, 22-lactone. Its characteristic feature is that the C8 or C9 side chains exhibit a lactone or lactol ring. However, lactone ring may be either five- or six-membered and may be fused with the carbocyclic moiety of the molecule via a C–C bond or through an O-bridge.

Many structures are also isolated from *W*. *somnifera*, which are the modifications or structural variants of withanolides which cause modifications to the carbocyclic backbone or side chains, such as withaferin A, withanolide A and D, withanone. These compounds are polyoxygenated and oxidize all C atoms of the steroid nucleus [5]. A new dimeric thiowithanolide also isolated from roots from *W. somnifera*, is named ashwagandhanolide as shown in Figure 3.

Withanolides act as a precursor hormone, in that it can be transformed into human physiology as needed [7]. *W. somnifera* is also sometimes referred to as Indian Ginseng because it structurally resembles ginsenosides (the active constituents of *Panax ginseng*). In addition, Withanolide A also possesses anti-dementia activities, and can thus be used for neuronal regeneration.

#### Withanolides Biosynthesis

Although the formation of withanolides in *W. somnifera* has not been fully elucidated, two pathways are reported:2-C-methyl-D-erythritol-4-phosphate (DOXP/MEP) pathway;The cytosolic mevalonate (MVA) pathway;The plastidic1-deoxy-D-xylulose 5-phosphate [52].

The biosynthesis starts with the formation of cholesterol by the action of acetyl co-enzyme A. Acetyl-co A is combined and converted into two types of mevalonic acid; only the R-type is used in triterpene biosynthesis, the S-type being inert in nature. Following the reduction of a C-atom, the R-type is transformed into isopentenyl pyrophosphate (IPP) which, together with its isomer 3,3-dimethyl allyl pyrophosphate (DMAPP), undergoes condensation to form geranyl pyrophosphate (GPP) and, after further condensation, farnesyl pyrophosphate (FPP). Two molecules of FPP condense in the presence of NADPH and squalene synthase to generate squalene which, after oxidation, forms a squalene 2,3-epoxide undergoes ring closure to form lanosterol, an intermediate of various steroidal triterpenoid structures. The formation of 24-methylenecholesterol has still not been determined, and it is unknown whether it is a precursor of steroidal lactones or not. Nevertheless, it has been observed that hydroxylation at C-22 and the δ-lactonization between C22 and C26 of 24-methylenecholestrol are part of the Withanolide pathway as shown in Figure 4.

### 3.2. Withaferins

4β,27-dihydroxy-1-oxo-5β, 6β-epoxywitha-2-24-dienolide, also known as Withaferin A, was first isolated from the leaf of South Asian *W. somnifera*, in 0.13–0.31% concentration of dry weight. Quantitative analysis of Indian chemotypes of *W. somnifera* by TLC densitometry showed that withaferin A was completely absent in whole plants, but present in leaves by 1.6% concentration [53].

Withaferin A and withanone (Figure 5) act as angiogenesis inhibitors and have anti-cancer properties [54,55]. Withaferin A also exhibits antibiotic activity due to its unsaturated lactone ring [56], and is applied to cure ulcers and skin carbuncles, inhibit tumor growth, exhibit radio-sensitizing effects in cancer treatment, and induce immune-suppressive activities on B- and T-lymphocytes. It also shows other activities such as antibacterial, antitumoral, anti-arthritic, and immunomodulating actions [43]. Withaferin A also ceases the growth of some bacteria and fungi [57].

### 3.3. Withanamides

Withanamides A-I (Figure 6) are purified from the methanolic extract of fruits of *W. somnifera*. Its structure contains a 5-hydroxy tryptamine (serotonin) base with hydroxy fatty amide and diglucosidic moieties. The difference between the structural variation in Withanamide A and Withanamide C is only in the fatty acid side chain, because Withanamide A has two double bonds in its side chain compared with Withanamide C.

Withanamide A and Withanamide C help neutralize the toxicity of β-amyloid protein (BAP) and protect the cells from cell death. Withanamide A is more effective in treating Alzheimer’s disease. It also exerts anti-oxidant properties, because it has the ability to inhibit lipid peroxidation (LPO). It inhibits tumor cell proliferation because it has selective cyclooxygenase (COX-1 and -2) enzymes [58,59].

### 3.4. Sitoindosides

Sitoindoside is a type of withanolide which contains glucose molecule at carbon 27, isolated plants of *W somnifera*. From its roots, two acyl glucosides sitoindoside VII and VIII, and two glycowithanoloids, sitoindoside IX and X are isolated. These phytochemicals possess anti-stress activity and also help in retaining the memory [47,48].

### 3.5. Alkaloids

As per the literature, in addition to steroidal compounds, many alkaloids are reported from the *W. somnifera* which are named somniferin (bitter in taste), somniferinine, somine, cuscohygrine, nicotine, pseudo-withanine, withananinine, tropine, 3α-trigloyloxytropane, anaferine, anhygrine, choline, withananine, and Withanine. The total alkaloid content present in roots varies from 0.13% to 0.31%, although it elevates up to 4.3% as was reported from different regions/countries, whereas there are five unidentified alkaloids present in leaves which yield approximately 0.09% (Figure 7). The differences in the yield of alkaloids may be because of some factors such as the method of isolation, environmental interactions, or variability in genotypes. These alkaloids show sedative and hypotensive effect [60].

### 3.6. Other

Other phytochemicals have been reported, such as Ipuranol, tannins, resin, fatty acids, organic acids, amino acids, reducing sugars, phytosterol, and saturated and unsaturated acids. The roots are reported to contain starch, reducing sugars, glycosides, dulcitol, withanicil, hentriacontane, and peroxidases, whereas in leaves, flavonoids, withanone, chlorogenic acid, calystegines (N-polyhydroxylated heterocyclic compounds), and tannin have been reported; in the berries, amino acids have been reported. Several amino acids have also been isolated from the root, including alanine, aspartic acid, cysteine, glycine, glutamic acid, and tyrosine [60].

## 4. Extraction of Phytochemicals

Extraction methods generally include traditional techniques such as digestion, infusion, maceration, decoction, infusion, infiltration, and Soxhlet extraction [61]. Microwave-assisted solvent extraction (MASE) techniques, ultrasound-assisted solvent extraction (UASE) techniques, and supercritical fluid extraction (SPE) techniques are the most common approaches used in the extraction of bioactive compounds of *Withania somnifera* [62,63]. The UASE and MASE methods have special advantages over the amount of solvent and extraction time and achieve higher yield than traditional extraction techniques. Fractionation and purification of the photochemical is performed using a variety of chromatographic techniques such as high-performance liquid chromatography (HPLC), gas chromatography (GC), paper chromatography, and thin-layer chromatography (TLC) [63]. The extraction of secondary metabolites from *Withania* spp. Is affected by many factors, such as the nature, its origin, degree of processing, moisture content, and particle size of the plant material. The results will differ according to the type of extraction technique and variables such as temperature and time. The phytochemicals in the extract may also be affected by the type of solvent and its concentration and polarity [64]. Thus, the extraction first depends on the type of plant material used, and whether it is fresh or dry, because different parts of the plants will vary in chemical composition according to their water content. The solvent choice depends on the polarity of target compounds. These should be non-toxic, evaporate at a low temperature, and not interfere with the assay. Many types of solvent, such as methanol, ethanol, ether, and water (traditionally used) [65], are used to extract anthocyanins, starches, tannins, saponins, terpenoids, polypeptides, lecithin, etc. (Table 6).

## 5. Biological Activity

*W. somnifera* possesses many vital biological activities because of the presence of valuable phytoconstituents in it. Some of its activities are as shown in Figure 8.

### 5.1. Anti-Stress and Anti-Anxiety/Psychotropic Activity

Withanolide glycosides show anti-stress properties and having positive effects on boosting memory [69]. They help treat stress-induced conditions such as gastric ulceration, cognitive deficit, irregular glucose homeostasis, sexual dysfunction, changes in plasma corticosterone levels, and immunosuppression [7].

*Withania somnifera* root and seed extracts show anti-stress activity. An in vivo study was performed in mice to determine the anti-stress activity of the plant. An alcoholic extract of roots and seeds was prepared in normal saline and a single dose (100 mg/kg) was administered intraperitoneally to 20–25 g mice to test their swimming performance in water at 28′–30′ C. The experiment showed a promising result; the swimming rate was almost double that of the control [70].

Research conducted at University of Texas Health Science Center showed that *Withania somnifera* extract slows down the brain by reducing neuron excitability and inhibiting nerve transmission, closely related to the primary inhibitor Gama-Aminobutyric acid (a neurotransmitter) [71].

Glycowithanolides of *Withania somnifera*, sitoindoside IX and sitoindoside X at a concentration of 50–200 mg/kg, and glycosides (sitoindosides VII and VIII) at a concentration of 50–200 mg/kg orally, showed significant anti-stress activity in albino mice and rats and resulted in increased memory retention [8].

Double-blind, placebo-controlled studies were performed to determine the anti-stress activity of *W. somnifera* in human trials. Sixty adults were randomly assigned to take 240 mg of a standardized ashwagandha extract (Shoden) once daily. The results were measured using the Depression, Anxiety, Stress Scale-21 (DASS-21), hormonal changes in cortisol, dehydroepiandrosterone-sulphate (DHEA-S), the Hamilton Anxiety Rating Scale (HAM-A), and testosterone levels. It was found that ashwagandha extract was associated with greater reductions than placebo for mean mHAM-A score, morning cortisol C-reactive protein, pulse rate, and blood pressure, and increased significantly for mean serum DHEAS and hemoglobin [72,73,74].

Studies suggest that the stress-relieving property of ashwagandha is due to modulation of the HPA axis. The HPA axis is an important hormonal response system which controls stress. This axis ensures that the body quickly responds to nerve-racking moments and returns to a normal state.

### 5.2. Anti-Spasmodic Activity

The alkaloids present in the plant have relaxant and antispasmodic effects against many spasmogens in uterine, intestinal, bronchial, tracheal, and blood vascular muscles. For example, Ashwagandholine exerts relaxant and antispasmodic effects [75].

### 5.3. Anti-Inflammatory Activity

*W. somnifera* possesses anti-inflammatory properties in both acute and chronic types of inflammation. Cultures of cartilage from patients with osteoarthritis and rheumatoid arthritis have been used to demonstrate its protective effects on chondroblasts [76]. Related effects on cytokines, transcription factors, and the suppression of nitric oxide (NO) have also been observed [77]. A decoction of root and leaf extracts containing alkaloids and withanolides proved effective against the denaturation of protein in vitro. Additionally, anti-inflammatory properties have been described for constituents such as withaferin A and 3-b-hydroxy-2,3-dihydrowithanolide F [78]. Thus, *W. somnifera* also exerts anti-arthritis effects and is used to treat osteoarthritis. It may be taken together with other supplements such as Articulin-F. This also works as an analgesic and helps to soothe the nerve system from pain.

### 5.4. Anti-Microbial Activity

*W. somnifera* has anti-microbial components in leaves and roots [79], which are effective against human pathogenic bacteria, fungi, and viruses [80,81]. The leaves of W. somnifera have anti-bacterial properties, and are effective against bacteria such as *Pseudomonas aeruginosa* and *Staphylococcus aureus.* Its antibacterial phytochemicals identified so far include withaferin A and 3-b-hydroxy-2,3-dihydrowithanolide F. The plant contains many alkaloids and other polar compounds with anti-bacterial activities which activate the immune system of the host [82]. Withaferin A inhibits the growth of various bacteria and pathogenic fungi; it is also active against *Micrococcus pyogenes*, partially active against *Bacillus subtilis*, and inhibits Ranikhet disease, vaccinia virus, and *Entamoeba histolytica* [43]. *W. somnifera* helps to provide protection against systemic *Aspergillus* infection by activating the macrophage function, which increases phagocytosis and the intracellular killing of peritoneal macrophages [83], while *W. somnifera* shows anti-fungal activity against *Helminthosporium sativum* [84]. Some glycoproteins derived from *W. Somnifera* inhibit the growth of phytopathogenic fungi (*Aspergillus flavus* and *Fusarium verticilloides*) by prohibiting their spore formation and the growth of hyphae [79]. Withaferin A has been found to inhibit infectious bursal disease virus [85], herpes simplex virus [86], HIV infection [87], and coronavirus [19].

### 5.5. Insecticidal Activity

Phytochemicals from *W. somnifera* have insecticidal activities and can protect from many insects such as *Callosobruchus chinensis*, *Sitophilus oryzae*, *Triboliumcastaeneum* [88,89], and *Spodoptera litura*, mosquito vectors such as *Anopheles stephensi*, *Aedes aegypti*, and *Culex quinquefasciatus* [90], and termites [91]. They also show herbicidal activity against the noxious weed *Parthenium hysterophorus*.

### 5.6. Anti-Diabetic Activity

*W. somnifera* acts as an anti-diabetic, reducing blood sugar and cholesterol levels [29]. It decreases streptozocin, which is particularly toxic to insulin-producing beta cells of the pancreas and induces hyperglycemia. The anti-diabetic effect of the plant may arise from pancreatic islet free radical scavenging activity. The hyperglycemic activity of streptozocin results from reduced pancreatic islet cell superoxide dismutase (SOD) activity, which leads to the accumulation of degenerative oxidative free radicals in islet beta cells [92].

### 5.7. Hepatoprotective Activity

Histopathological studies confirm that the alcoholic extract of *W. somnifera* leaves inhibits CCl_4_-induced alterations in transaminase activity and pentobarbitone sleeping time, indicating hepatoprotective properties [93]. This also inhibits ochratoxin A, which causes liver inflammation and suppresses macrophage chemotaxis [83].

### 5.8. Anticarcinogenic Activity

Various studies have demonstrated the anticarcinogenic effects of *W. somnifera* secondary metabolites in animals and cell cultures, especially withaferin A and withanone [94]. Due to a broad spectrum of cytotoxic and tumor-sensitizing actions, the plant has potential applications in novel complementary therapies for integrative oncology. Effects against the HL-60 leukemia cell line, myeloid leukemia, and bladder, breast [95], prostate, colon, kidney, gastric, and lung cancer, have been described [96]. The mechanisms of *W. somnifera*-derived anticarcinogenic activity include antiproliferative effects [97], apoptosis [98], radio-sensitization, mitotic arrest, antiangiogenics, and the enhancement of cell defense mechanisms [99]. Leaf extracts have been used for the selective killing of cancer cells. Withaferin A is more effective than doxorubicin in inhibiting breast and colon cancer cell growth [58]. Due to these functions, *W. somnifera* bioactive agents have the potential to fight cancer by reducing tumor cells [100,101]. *W. somnifera* displays activity against urethane-induced lung tumors in mice and radio-sensitizing actions. Withaferin A, 3-hydroxy-2,3-dihydrowithanolide F, and withanolides D and E all showed anti-tumor activity in vitro against human epidermoid nasopharynx carcinoma and in vivo against Ehrlich ascites carcinoma, sarcoma, and mammary adenocarcinoma. They also acted as mitotic poison against human larynx carcinoma cells at the metaphase. The effect of Withaferin A also increased with the help of Methyl-thiodeacetyl colchicine. *W. somnifera* has many phytochemicals with unsaturated lactones in the side chain attached to the allelic primary alcohol group at C25 and highly oxygenated rings at the other end of the molecule, which may afford them cariostatic properties [43]. This also helps to reverse the harmful effects of urethane on lymphocytes, improve the leukocyte count, increase body weight, and reduce mortality [70]. Fibroid tumors of the uterus are also treated and help to reduce uterine bleeding [102].

*Withania somnifera* exhibits anticancerous property due to its phytochemicals (Withanolide and withaferin). Withanolide suppresses the inducible and constitutive expression of the “NF-kappa B” signaling pathway involved in extensive cancer development [103]. In addition, Withaferin induces apoptosis in tumor cells and prevents their spreading [104].

### 5.9. Cardiotonic Activity

*W. somnifera* helps to purify blood. It is used to treat heart weaknesses and blood disorders [29]. The effects of digoxin (a drug used to treat irregular heartbeat and heart failure) are similar to this plant. *W. somnifera* helps to reduce blood pressure by blocking the action of autonomic ganglia and exerting myocardial depressant, as well as positive inotropic and chronotropic effects [70]. *W. somnifera* also exhibits hypercholesteremic, hypolipidemic, and anti-atherogenic activities. It works against atherogenesis and vascular intimal damage. It inhibits lipid peroxidation, platelet aggregation, delays the plasma re-calcification time, and improves the release of the lipoprotein lipase enzyme. It reduces body weight and increases high-density lipoprotein cholesterol levels [105]. The root powder has anti-aging effects, as it increases red blood cells and hair melanin, and decreases serum cholesterol [58,71,78].

### 5.10. Central Nervous System (CNS)

*W. somnifera* helps to treat depressive disorders and exhibits a tranquilizing effect. Animal studies suggest that the plant components affect the central nervous system, including the modulation of acetylcholinesterase and butyrylcholinesterase activity, the inhibition of calcium ion influx, blockade of gamma-aminobutyric acid receptors, modulation of 5-HT 1 and 5-HT 2 receptors, and the regeneration of neurites. [70,106] Glycowithanolides, withaferin-A, and sitoindosides VII–X obtained from the roots reduce the effect of ibotenic acid, which induces cognitive defects. Roots of *W. somnifera* are used as a sedative, a treatment for insomnia, and nervine tonic. They are useful in cases of fainting, dizziness, and insomnia. A well-known adaptogen, *W. somnifera* helps to protect the body from damaging effects and promotes normal physiological functioning. It also works on the neuroendocrine system [7] and is thought to help the body cope with stress, boost the whole nervous system, and improve attention and concentration power. They are used in the treatment of Parkinson’s, Alzheimer’s, neurotic atrophy, synaptic loss, and other neuro-degenerative diseases because of its radical quenching properties [107].

### 5.11. Thyrotropic Effect (Hypothyroidism)

Research has revealed that *W. somnifera* has thyrotropic effects [108]. Its constituents decrease thyroid activity and lipid peroxidation in the liver homogenate and affect cellular antioxidant systems by increasing catalase activity while T3 levels remain constant. Additionally, urine volume and sodium are increased, and serum cholesterol and triglycerides are reduced.

### 5.12. Immunomodulatory Activity

*W. somnifera* works as an immunomodulator in indigenous medicines [75], and its phytochemicals, such as withaferin A, 3-b-hydroxy-2,3-dihydrowithanolide F, glycowithanolides and sitoindosides IX and X have proven immunomodulatory properties. The activities are similar to some immunosuppressive drugs such as cyclophosphamide, azathioprine, and prednisolone. For example, spleen cell proliferation is inhibited by withanolides. Murine macrophages and phagocytosis are activated by withanolide glycosides, which also increase lysosomal enzymatic activity secreted by the macrophages. *W. somnifera* helps to increase hemoglobin concentration, white and red blood cells, platelets, and body weight, and reduces leukopenia induced by cyclophosphamide (CTX) or sublethal doses of gamma radiation. Moreover, it helps to slow hypersensitivity reactions. It increases the cytotoxic effect of nitric oxide synthase of macrophages, which works against microorganisms and tumor cells [70,71,108].

### 5.13. Antioxidant Activity

*W. somnifera* contains many powerful antioxidant phytochemicals such as polyphenols, [16] sitoindosides VII–X, withaferin A, and glycowithanolides [70,71]. The brains and nervous systems of humans are rich in lipids and iron; therefore, they are particularly susceptible to damage by reactive oxygen species (ROS). Our brain uses approximately 20% of the total oxygen supply of the organism, and free radical damage to the nervous system causes heavy neuronal losses [69].

### 5.14. Anti-Peroxidative Activity

Root extracts of *W. somnifera* regulate the effects of lead toxicity and the lipid peroxidative process in liver and kidney tissues [109]. This treatment enhances hepatic and renal lipid peroxidation, decreases lipid peroxidation, and increases the activities of antioxidant enzymes such as superoxide dismutase (SOD) and catalase (CAT), thereby helping to maintain normal oxidative status of the tissues [110].

### 5.15. Male Infertility

*W. somnifera* helps in maintaining fertility in men by improving the quality of sperm but not the sperm count [29,70]. The root extract, taken as an aphrodisiac, increases the sex hormones testosterone and cortisol [111]. It also increases semen volume, sperm concentration, and sperm motility [31,112]. Overall, it works against stress and infertility [113]. *W. somnifera* is also known as *sukrala*, i.e., it helps to increase semen, and is regarded as an aphrodisiac. The powdered plant mixed with ghee, sugar and milk is taken to treat puerperal backache and leucorrhea caused by endometritis [28,29].

### 5.16. Anti-COVID-19 Activity

COVID-19 (novel coronavirus infectious disease) is caused by SARS-CoV-2, a positive-sense single-stranded RNA virus whose genome encodes four major proteins: a nucleocapsid, envelope, spike, and membrane [114,115]. By blocking the main virus, a protease or 3-chymotrypsin-like protease, Withanoside V, produced by *W. Somnifera*, delays maturation [19,78,116,117].

## 6. Applications in Medical and Food Systems

*W. somnifera* is rapidly advancing and is used for wide range of applications in medicinal and food sectors. Its potential is widely studied and can be used tradionally to treat number of health problems. The Table 7 shows some of its functions in different modes of uses which are as follows.

### 6.1. Medical Applications

*Withania somnifera* have wide range of pharmacological activities and numerous medicinally important products in the form of capsules, tablets, gummy, and powder have been developed by different manufactures, as shown in following Table 8.

### 6.2. Food Applications

The clinical role of *Withania somnifera* has been thoroughly explored, but little knowledge is available concerning food applications [134]. Its leaves and roots are good sources of dietary fiber [135]. Leaves and roots of incorporated products of *W. somnifera* can be used as extrudates, juices and beverages, sweet products, bakery and cereal products, dairy products, etc. as shown in Table 9 [134,136]. Numerous therapeutically important compounds have been observed to possess various biological activities. These compounds could play major roles in the dietary system.

#### 6.2.1. Juices and Beverages

*Withania*-*somnifera*-fortified beverage blends could be utilized for its nutritional quality in functional fruit beverages and the developed stored product was stable and acceptable for 90 days at room temperature [137].

#### 6.2.2. Sweets Products

Shrikhand sweets are said to contain 0.5–0.6% ashwagandha powder and improve stability and self-life up to 52 days at refrigerator temperature [138].

#### 6.2.3. Bakery and Cereal Products

Ashwagandha powders, including many products, are evaluated at different concentrations, such as 5% in Namakpara, Pap Chakal, and Muruk, and up to 10% in Missy Roti and Chutney powders. Containing up to 5% of *Withania somnifera* root powder, biscuits are rich in energy, minerals, fiber, and protein, enhancing their medicinal benefits [139]. Ashwagandha root powder at 2% in baked goods such as Thepla and breads reduces diabetes [140].

#### 6.2.4. Dairy Products

Pawar et al. (2014) [141] reported that ashwagandha powder, vidarikand, and shatavari containing ghee (fat) have stronger antioxidant properties than traditionally prepared ghee.

**Table 9 molecules-28-01208-t009:** Different manufactures marketing *Withania*-based formulations in food sector.

Product Name	Ingredient(s)	Functions	Manufactures	Reference
Ashwagandha Leaf Juice	Ashwagandha leaf	Immunity booster Helps in weight management Eases stress	Axiom	[142]
Ashwagandha Ginger Juice	Ashwagandha root Ginger	Helps boost strength and stamina	Baidyanath Vansaar	[143]
Artho Sure Juice	Ashwagandha, Guduchi, Kutki, and other herbs	Relieve joint pain	Kapiva	[144]
Sweet Himalayan Green Tea	Ashwagandha, Shatavari, Stevia, turmeric, and cardamom	Eases stress	Navvayd	[145]
Chyawanprash	Ashwagandha Amla giloy shatavari pippali mulethi	Helps build Strength and Stamina	Dabur	[146]
Moon Latte	Ashwagandha with curcumin, nutmeg, cinnamon Shankhpushpi black pepper	Helps in relaxing the body	Nature’s Island	[147]
Hot chocolate	Ashwagandha powder Cocoa powder	Boosts immunity Strengthens bones Detoxify mind and body	La rama	[148]
Cocoa hazelnut spread	Hazelnuts, ashwagandha, cocoa powder, soya bean, and brown sugar	Healthy for the body	Amaara	[149]
Chocolate spread	Ashwagandha, sankhpushpi, brahmi, peanuts, almonds, cashews, hazelnuts	Brain booster	Kids and teens	[150]

## 7. Future Perspectives

*W. somnifera* could be used as potential biomolecules for developing pharmaceutical products and formulations. In addition to developing a fine rational management for *W. somnifera* as a miraculous plant in the future, it may be useful for inspecting process and prospective of its remedial uses and food packages of *W. somnifera*.

## 8. Conclusions

In summary, *W. somnifera* has antiquity uses to treat various diseases. This review indicates that many pharmacologically important compounds are present in it which performs crucial role in natural medicinal systems, and also lays foundation for next generation.

*W. Somnifera* (Ashwagandha), a potential medicinal herb, has promising therapeutic and pharmacological properties due to its diverse phytochemicals. Many studies on this medicinal herb have shown antidiabetic, antistress, anti-inflammatory, anti-cancerous, anti-COVID-19, immunomodulator, antimicrobial, and hepatoprotective activity. Importantly, Withaferins induces apoptosis in tumor cells and prevents their spreading. Glycoproteins derived from *W. Somnifera* inhibit the growth of phytopathogenic fungi. The literature reveals that anti-anxiety properties are mainly because of glycowithanolides and glycosides.

Value-added products from *Withania* spp. include capsules, herbal tea, syrups, powder, root powder, root extract, strips, juice, tablets, oil, tonic, decoction, poultice, herbal beer, traditional drugs, and health drinks. Furthermore, there should be assessments on the quantitative improvement and exploration of its value in commercial sectors and herbal sectors. Thus, this medicinal plant is a good target for future studies in the fields of phytomedicines, drug development, and herbal sectors.

In a nutshell, in vivo and in vitro research should be conducted to determine the modes of action and molecular mechanism of phytochemicals for drug development and design. Phytochemicals exhibiting antiviral activity will also be potential candidates for developing antiviral medicines against COVID-19. However, more clinical trials should be conducted to prove its therapeutic and pharmacological properties.

In this review, we have compiled the detailed biochemical profiling of phytochemicals *W. Somnifera*, their extraction, and its value-added products and therapeutic and pharmacological properties. In summary, many pharmacologically important compounds are present, as illustrated in natural medicinal systems, as well as laying the foundation for the next generation.

## Figures and Tables

**Figure 1 molecules-28-01208-f001:**
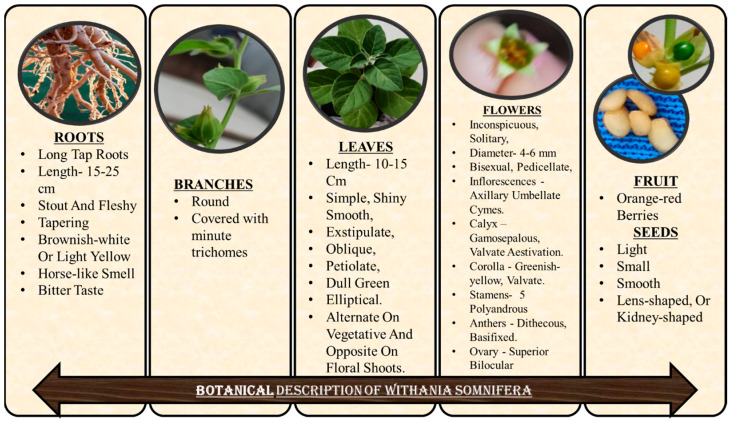
Botanical description of *W. somnifera*.

**Figure 2 molecules-28-01208-f002:**
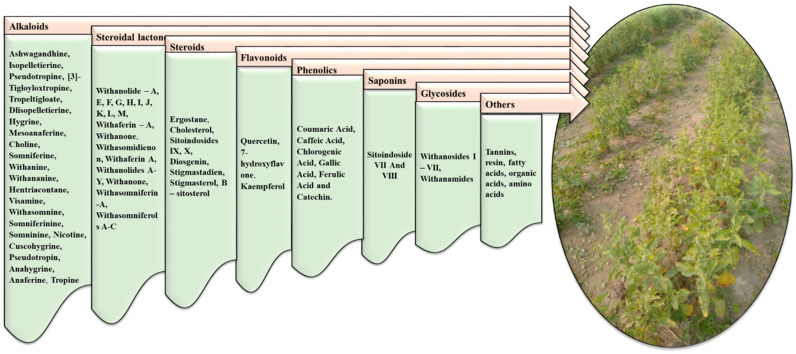
Various classes of bioactive compounds present in *Withania somnifera*.

**Figure 3 molecules-28-01208-f003:**
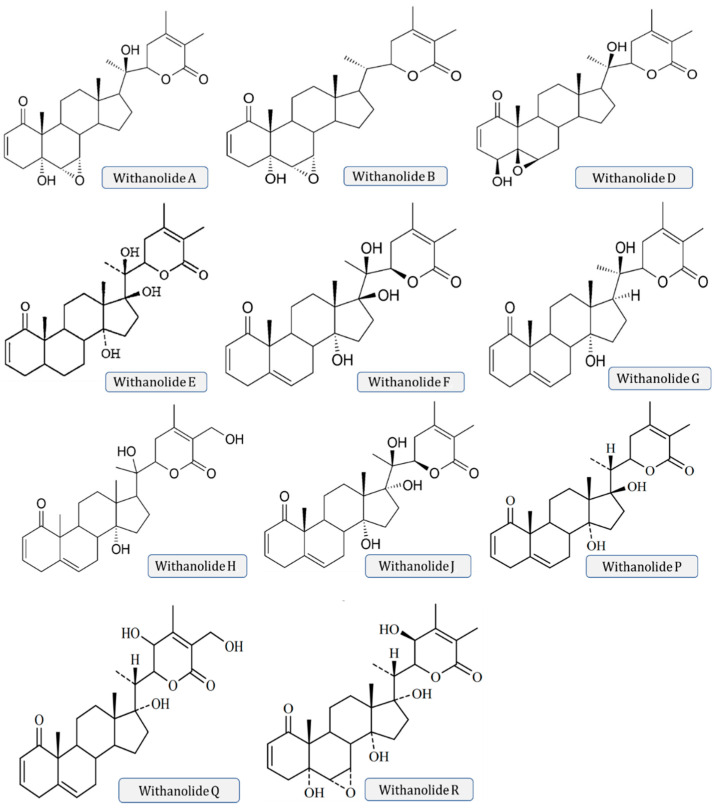
Various chemical structures of different withanolides.

**Figure 4 molecules-28-01208-f004:**
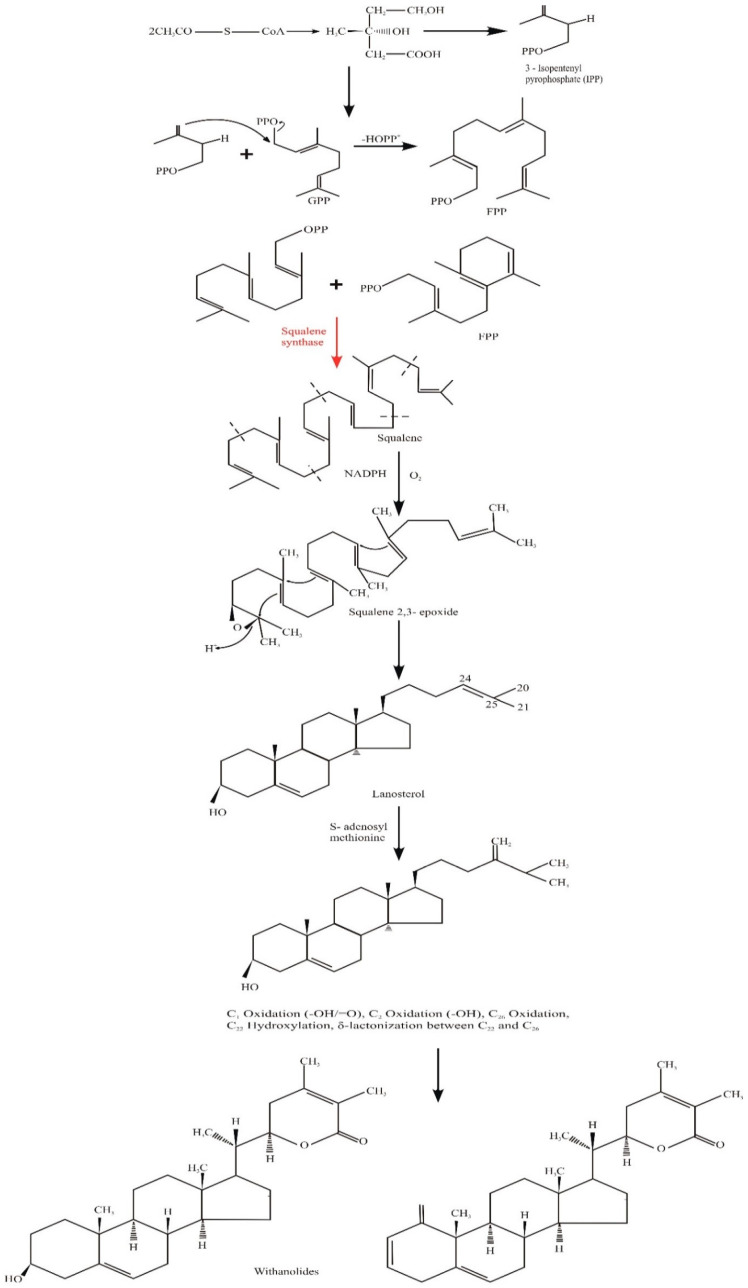
Pathway of the biosynthesis of withanolides in *W. somnifera* [5].

**Figure 5 molecules-28-01208-f005:**
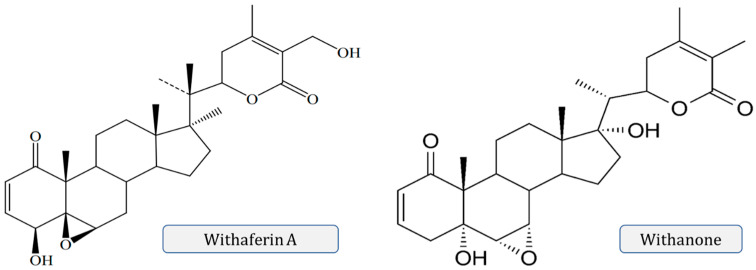
Chemical structure of withaferin A and withanone.

**Figure 6 molecules-28-01208-f006:**
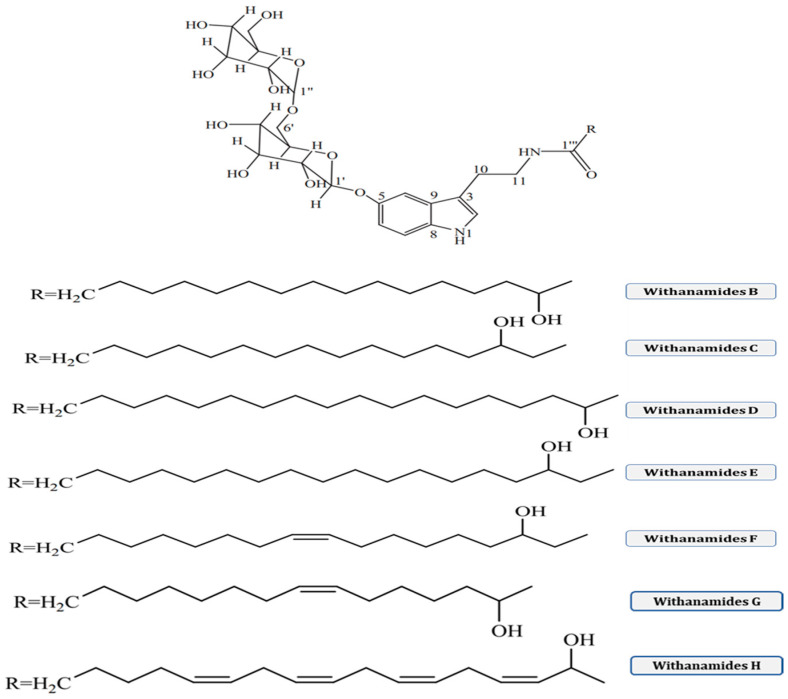
Various chemical structures of different withanamides.

**Figure 7 molecules-28-01208-f007:**
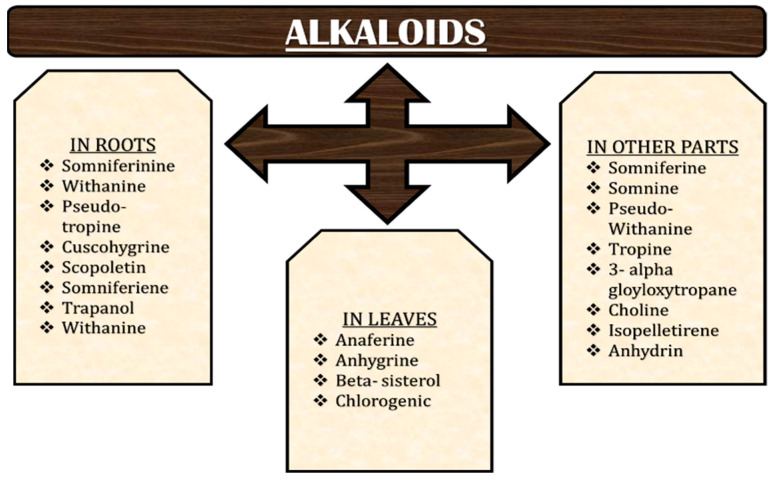
Various alkaloids present in different parts of *W. somnifera*.

**Figure 8 molecules-28-01208-f008:**
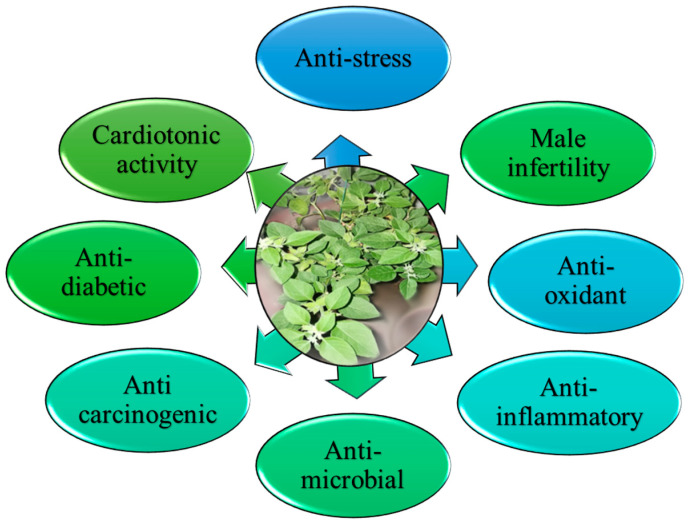
*W. somnifera* possesses various biological activities.

**Table 1 molecules-28-01208-t001:** The different biological species of the Withania genus, their availability throughout the world, and different bioactive compound.

Biological Genus	Availability	Bioactive Compound	Uses	References
*Withania somnifera*	Subtropical regions of India, Africa, Asia, Australia, and Europe	Alkaloids, phenols, flavonoids, saponins, tannins, carbohydrates, steroidal lactones, β-sitosterol, scopoletin, sitoindosides, somniferiene, somniferinine, pseudotropine, anaferine, anahygrine, cysteine, chlorogenic acid, cuscohygrine, withanine, withaferine, withanolides, withananine, tropanol, 6,7β-Epoxywithanon, 14-α-hydroxywithanone, etc.	Antimicrobial, anticancer, anti-inflammatory, antistress, antidiabetic, antioxidant, and reproductive impotence	[5,7,16,17,18,19,20,21]
*Withania adpressa*	North Africa, Morocco, Algeria, Mediterranean Basin, and India	Wadpresine, withanolide Fand J, coagulin L, nicotiflorin	Antiproliferative immunomodulatory	[13,22]
*Withania coagulans*	Mediterranean region, North Africa to South Asia, Iran, Pakistan, Afghanistan, and East India	Protein, fat, fiber, carbohydrates, minerals, amino acid, fatty acid, phenols flavonoids, tannins, withanolides, coagulans, coagulanolide, coagulins, anthocyanins, etc.	Antifungal, anti-cytotoxic, antidiabetic, hypolipidemic, neuroprotective, anti-inflammatory, anticancer, anthelmintic, and antioxidant	[14,23,24]
*Withania frutescens*	North Africa, South and Western Asia and Southern Europe	Polyphenols, tannins, mucilage, terpenoids, flavonoids, and saponins	Antifungal, antibacterial, anti-inflammatory, antidiabetic, and wound healing	[15,25,26]
*Withania aristata*	North Africa and the Mediterranean Basin	Withanolide A, withanoside IV, withaferin A, 27-hydroxywithanone, withanoside V, withastramonolide withanone, withanolide, and physagulin	Anticholinesterase, anti-cancereous, anti-inflammatory, analgesic, and antioxidant properties	[24]

**Table 2 molecules-28-01208-t002:** Different biological compounds present in leaves and their activities.

Name of Phytoconstituents	Biological Activities	References
Free amino acids, 0.09% unidentified alkaloids, chlorogenic acid, condensed tannins, flavonoids, glucose, glycosides, N-heterocyclic compounds, withaferin A (steroidal lactone), twelve withanolide, withanone	Anti-helmantic, anti-inflammatory, anti-stress, anti-anxiety, anti-carcinogenic, anti-microbial, and anti-oxidant	[5,28,29]

**Table 3 molecules-28-01208-t003:** Different biological compounds present in roots and their activities.

Name of Phytoconstituents	Biological Activities	References
Alkaloids (0.13–0.31%), alanine, anahygrine, anahydrine, anaferine, aspartic acid, chlorogenic acid, choline, condensed tannins, cuscohygrine, cysteine, dulcitol, flavonoids, free amino acids, glucose, glutamic acid, glycosides, glycine, starch, volatile oil, reducing sugars, hentriacontane, withanicil, withaniol, steroidal lactones, iron, somniferine, somnine, somniferinine, withananine, withanine, withasomnine, tropine, psuedotropine, 3-α-gloyloxytropane, isopelletierine, tyrosine, tryptophan, visamine, withaferin A, withanolide A, sitoindoside VII, VIII, IX, and X.	Anti-inflammatory, anti-insecticides, improves fertility, improves memory, cerebellar ataxia, cardiotonic, anti-oxidant, and anti-peroxidative	[30,31]

**Table 4 molecules-28-01208-t004:** Different biological compounds present in fruits and flowers, and their activities.

Parts of Plant	Name of Phytoconstituents	Biological Activities	References
Fruits and flowers	Amino acids, aspartic acid, alanine, chamase, condensed tannins, cystine, flavonoids, glutamic acid, glycine, hydroxyproline, isopsoralen, peroxidases, proteolytic enzyme, Psoralen, proline, tyrosine, and valine.	Anti-microbial, milk thickening agent, and treats respiratory problems	[34,35,36]

**Table 5 molecules-28-01208-t005:** Various chemical constituents and their derivatives found in *Withania somnifera*.

Bioactive Compounds	Part of Plant	Derivatives	Yield	Uses	References
Alkaloids	Leaves Roots Stems	Ashwagandhine, Isopelletierine, Pseudotropine, [3]-Tigloyloxtropine, Tropeltigloate, Dlisopelletierine, Hygrine, Mesoanaferine, Choline, Somniferine, Withanine, Withananine, Hentriacontane, Visamine, Withasomnine, Somniferinine, Somninine, Nicotine, Cuscohygrine, Pseudotropin, Anahygrine, Anaferine, Tropine	0.013–119 mg/gm	Anti-microbial, sedative, relaxant, anti-spasmodic; used to treat tumors, nocturnal leg cramps, diarrhea, and psychiatric palpitation	[42,43,44]
Tannins	Roots Leaves Fruits Flowers	N/A	9.46–23.02 mg/gm	Anti-fungal, anti-biotic, anti-inflammatory, analgesic, astringent, and wound healing	[36,42,43,44]
Glycosides	Roots Stems	Withanosides I–VII, Withanamides	19.01–122.26 mg/gm	Sedative, muscle relaxant, diuretic, anti-microbial, anti-inflammatory, and anti-cancer	[42,43,44]
Saponins	Roots Berries	Sitoindoside VII And VIII	3.02–5.26 µg/gm	Anti-inflammatory, anti-hepatotonic, hypoglycemic, anti-microbial and antiviral; used in detergents and molluscicides	[43,44]
Flavonoids	Roots Stems	Quercetin, 7-hydroxyflavone Kaempferol	5.1–80.23 mg/gm	Anticancer, antioxidant antimicrobial strengthen capillary walls, improves blood cholesterol levels as well as reduces the risk of cancer, osteoporosis, and coronary heart diseases	[36,42,43,44,45]
Steroids	Roots	β-sitosterol, Cholesterol, Diosgenin, Ergostane, Sitoindosides IX, X, Stigmastadien, Stigmasterol	0.1–0.45 mg/gm	Aphrodisiac, reduces cholesterol levels, affects the immune system, and tumor cells	[43,44]
Steroidal lactones	Leaves Roots	Withaferin–A, Withanone Withasomidienone, Withanolides A-Y, Withasomniferine, Withasomniferols A-C	0.01–0.8 mg/gm	Anti-inflammatory, anti-anxiety, and improves fertility	[5,36,46,47]
Phenolic	Roots Stems	Coumaric acid, caffeic acid, chlorogenic acid, gallic acid, ferulic acid, and catechin.	90.325 mg/gm	Anti-inflammatory, anti-oxidants, anti-cancer, and anti-septic	[43,44]

**Table 6 molecules-28-01208-t006:** Extraction from different solvents shows the presence and absence of different phytochemicals [27,47,66,67,68].

Phytochemicals	Test	Plant Parts	Extraction Methods								
			Ethanol	Methanol	Chloroform	Aqueous	Ethyl Acetate	Hexane	Benzene	Petroleum ether	Acetone
Alkaloids	Mayer test Wagner test	Roots	+	+	-	+	+	-	+	+	+
		Leaves	+	+	+	-	+	-	+	+	+
		Stem bark	+	+	+	-	+	-	+	+	+
Tannins	Lead acetate Test 5% FeCl_3_ Test	Roots	+	+	+	+	+	-	+	+	+
		Leaves	-	-	+	-	+	+	-	-	+
		Stem bark	-	-	+	-	+	+	-	-	+
Glycosides	Benedict’s test Fehling’s test	Roots	+	+	-	+	+	+	+	+	+
		Leaves	-	+	-	-	+	-	+	-	+
		Stem bark	-	-	-	-	+	-	+	-	+
Saponins	Foam test	Roots	-	-	-	+	+	-	-	+	+
		Leaves	+	+	+	-	+	-	-	-	+
		Stem bark	+	+	+	-	+	-	-	-	+
Flavonoids	Lead acetate test Sulfuric acid test 5% FeCl_3_ test NaOH test	Roots	-	+	+	+	+	+	+	+	+
		Leaves	-	+	+	+	+	+	+	+	+
		Stem bark	-	+	+	+	+	+	+	+	+
Steroids	Salkowski Liebermann–Burchard test	Roots	+	+	+	+	+	-	+	-	+
		Leaves	+	+	-	+	+	+	+	-	+
		Stem barks	+	+	-	+	+	+	+	-	+
Phenolic	5% FeCl_3_ test Gelatin test	Roots	+	+	+	-	+	-	+	-	-
		Leaves	+	-	+	-	+	+	-	+	+
		Stem bark	+	+	+	-	+	-	-	-	+
Cardiac glycosides	Keller–Killiani test Fehling’s test Sodium nitroprusside test	Roots	+	-	-	-	+	-	+	+	+
		Leaves	+	+	-	-	+	+	+	-	+
		Stem barks	+	+	-	-	+	+	+	-	+
Terpenoids	Liebermann–Burchard test	Roots	+	+	+	+	+	+	-	N/A	+
		Leaves	+	+	-	-	+	+	+	+	+
		Stem barks	+	+	-	-	+	+	+	+	+
Proteins	Ninhydrin test Xanthoproteic test	Roots	-	-	-	-	+	-	-	-	+
		Leaves	-	-	+	+	+	-	+	+	+
		Stem bark	-	-	+	+	+	-	+	+	+

“+” shows presence and “-” shows absence.

**Table 7 molecules-28-01208-t007:** Various modes of uses of different parts of plant.

Scheme	Plant Parts	Mode Of Uses	Functions	Reference
1.	Leaves	Paste	On enlarged cervical and other glands, reduce edema and pain	[28,29,37,78,118]
			Applied to burns and wounds	
			Used as a sunscreen	
			Treat skin diseases, carbuncles, syphilitic sores, tumors, ulcers, tubercular inflammation, and also helps to heal ulcers and bronchitis	
		Juice	As eardrops, decreases ear discharge problems	[28,29]
			Treating conjunctivitis and anthrax pustules	
		Fodder	Given to livestock	[5]
		Powder	Applied to burns and wounds	[28]
		Warm Leaves	Provides relief against eye illnesses	[29]
		Infusions	Treat fevers	[5,8,78]
		Decoctions	Treat sore eyes, boils, swellings in limbs and piles	[29,118]
		Ointment	Serves as an insecticide to kill lice infesting the body, and is also useful to treat bed sores, wounds, and cuts	[29,37]
2.	Roots	Milk boiled with fresh roots	To leach out undesirable waste from the body	[28,37]
		Paste	Treats swellings and ulcers	[28,37]
		Tonic	Treat sexual deficiency in males	[29,37,119]
		Powder	To slow the growth of tumors, and improve male sperm count	[37]
		Decoctions	To treat sterility in females, chest complaints, scrofula, and colds	[5,37]
		Black ashes	To heal blisters and swellings	[5]
		Others	To treat rheumatism, loss of appetite, cough, dropsy, dyspepsia, joint pain, leukoderma (a skin disorder), inflammation, nervous breakdown, senility, debility, loss of memory, goiter, lower back pain, sciatica, boost immunity, increase white blood cells in the body, treat insomnia, the regulation of blood sugar, and reduction in bad cholesterol	[5,8,29,37]
3.	Seeds	N/A	To expel parasites out of the body, to treat chest problems, to thicken milk	[29,37,78]
4.	Berries	N/A	To coagulate milk Sedative, emetic, blood-purifying, febrifuge, and diuretic Treats dyspepsia, to promote growth, treats liver problems, asthma, and biliousness	[5,8,29,37]
5.	Barks	Decoction	To treat asthma and bed sores	[5,29]
6.	Twigs	Chewed Non-Fibrous Twigs	To clean teeth	[5]
		Steaming	Provides relief from toothache	

**Table 8 molecules-28-01208-t008:** Different manufactures marketing *Withania*-based formulations in the medical sector.

Form	Product Name	Ingredient(s)	Functions	Manufacturers	Reference
Capsules	Kapiva Capsules Ashwagandha	Ashwagandha extract *Withania somnifera* (root)	Vital immunity booster, nourishes and strengthens the nervous system, and increase stamina	Kapiva	[120]
	KSM-66 Ashwagandha	KSM-66^®^ Ashwagandha root extract	Reduced anxiety, better immunity, better sleep quality, and improves male fertility	Better Veda	[121]
	Ashwagandha Veg Capsule	Organic Ashwagandha root extract	Stress, anxiety, immune deficiency, low energy, chronic fatigue, general exhaustion, cold flu, premature aging, and depression	Organic India	[122]
	HK Vitals Ashwagandha Powder	Ashwagandha extract *Withania somnifera* (root)	Focus, stress relief, and muscle strength	Health Kart	[123]
	Ashwagandha	*Withania somnifera* powder	Boost energy, muscle recovery, anxiety relief, and brain health	Himalayan Organics	[124]
Tablet	Ashwagandha General Wellness	Ashwagandha (*Withania somnifera)* root extract	Stress relief, overall health, and fatigue	Himalayan	[125]
	Ashwagandha	Ashwagandha (*Withania somnifera)* root extract	Immunity booster, immune modulator, relieve mental stress, body rejuvenated, and rich in anti-oxidants	Dabur	[126]
	Ashwagandha	Root extract	General health, anxiety and stress relief, energy, and endurance	True Basics	[127]
	Ashwagandha	*Withania somnifera* extract (3.5% Withanolides), Piper nignum extract (Piperine 95%), Vitamin E (d-alpha tocophery acetate), B-complex (Vitamin B6, Vitamin B9, Vitamin B12)	Anxiety relief	Neuherbs	[128]
	Ashwagandha (*Withania somnifera*)	Ashwagandha (*Withania somnifera)* Root powder	Combat cancer, reduce blood sugar levels, cortisol levels, increase fertility and boosts testosterone, and reduce inflammation	Dhootpapeshwar	[129]
Powder	Ashwagandha	*Withania somnifera* powder	Energy management, improves sleep, reduce stress and anxiety, muscle mass and strength	Jain	[130]
	100% Organic Ashwagandha Powder	*Withania somnifera* Root powder	Anxiety relief	Carmel Organics	[131]
	Organic Ashwagandha Powder	Root powder	Immune support	Just Jaivik	[132]
Gummy	Ashwagandha Gummies	Ashwagandha, Vitamin D	Promotes immunity, bone and brain health, strength endurance, and mental well being	Man Matters	[133]

## Data Availability

Not applicable.

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
