# Peer review of "Biodiversity, Biochemical Profiling, and Pharmaco-Commercial Applications of Withania somnifera: A Review"

_molecules, 2023, doi:10.3390/molecules28031208_

Round 1

Reviewer 1 Report

The relevant content on speceis distributions should be add to this manuscript. 

Author Response

Respected Reviewer,

As per your valuable suggestions, corrections have been made in page number 3 which shows the content related to different species and its distributions.

Reviewer 2 Report

REVIEW REPORT

Hello worthy editor and authors,

The interesting manuscript titled as “Biodiversity, biochemical profiling, and pharmaco-commercial applications of Withania somnifera: A review” is a significant contribution into the field of medicinal plants and phytochemistry. The current review article is a comprehensive package which focuses on the traditional uses, botany, phytochemistry, and pharmacological progress of the bioactive constituents of Withania somnifera. The authors have done comprehensive review literature and reported that at least 62 major and 48 minor primary and secondary metabolites are present in the Withania somnifera leaves, and 29 among these are found in leaves and roots of Withania somnifera. The manuscript is well written and theory is easy to understand and I think it will attract wide readership. However, before the paper can be considered for publication/acceptance, it is necessary for authors to undertake minor revisions. I hope that this article will appeal wide readership attention. However, I do find some minor shortcomings in the article that the author need to correct before publication. The manuscript is well written but I would recommend revision and improvement of certain sentences/statements. I found this article insightful, applied and quite informative and recommend it for publication with minor revision.

§  The abstract is well written and has excellently discussed and concluded the different sections of the manuscript. However, it would be highly appreciated if authors incorporate sentence about the future perspectives of the research work conducted from present study.

§  The introduction section is quite informative and focused. Authors have comprehensively discussed the notion behind present research work. Authors are advised once again to double check grammar, sentence structure etc, if there any deficiency, fix them accordingly.

§  There is lack of cohesion among different paragraphs. Carefully review the manuscript and bring cohesion among different paragraphs.

·         Support your statement with updated references if available.

§  Cross check all the references and strictly follow author guidelines.

Author Response

Respected Reviewer, 

As per your valuable suggestions, Grammer and other corrections have been made and also written future perspectives on page number 20. 

Reviewer 3 Report

My comments:

The topic name of the article  should be appropriately changed to Biodiversity, biochemical profiling, and pharma-pharmaceutical development and commercialization  of Withania somnifera: 

1 Please see some of the grammar errors etc.

2. In Section 6. There are 3 Tables;

 Section 6. Applications in medical and food systems.

Table 6.1. Various modes of use of different parts of the plant ( Gramma errors)

Table 6.2 Different manufacturers marketing Withania-based formulations in the medical sector

Table 6.3 Different manufacturers marketing Withania-based formulations in the food sector

Tables 6.2 and 6.3 do not have Tables' Names and References ..Why?

3. The review is quite too general, it has no important novel ideas or critical discussion.

4. See more highlighted comments in the file attached at the end.

Author Response

Respected Reviewer, 

As per your valuable suggestions, corrections have been made as follows:

Title have been changed as per your suggestions.

Grammatical errors have been removed.

Table name and references have been corrected. 

Other highlighted comments also corrected.